# Are Perfectionists Always Dissatisfied with Life? An Empirical Study from the Perspective of Self-Determination Theory and Perceived Control

**DOI:** 10.3390/bs12110440

**Published:** 2022-11-10

**Authors:** Qipeng Liu, Xiaoyun Zhao, Weidi Liu

**Affiliations:** School of Education, Huaibei Normal University, Huaibei 235000, China

**Keywords:** perfectionism, perceived control, life satisfaction, basic psychological needs, self-determination theory

## Abstract

Compared to non-perfectionists, perfectionists may not be satisfied with the growing needs in their lives to the same extent. To test whether perfectionists are dissatisfied with their lives, we investigated whether trait perfectionism attenuates the relationship between basic psychological needs, perceived control, and life satisfaction. A total of 574 college students self-reported basic psychological needs, perceived control, life satisfaction, and perfectionistic strivings and concerns, with a mean age of 19.53 (SD = 1.61), including 299 women and 275 men. A correlation analysis showed that perfectionistic strivings were significantly positively related to life satisfaction, while perfectionistic concerns were significantly negatively related to life satisfaction. The moderation analysis showed that perfectionistic strivings not only moderated the relationship between basic psychological needs and life satisfaction but also moderated the relationship between perceived control and life satisfaction. Individuals with high perfectionistic strivings generally reported high levels of life satisfaction. Perfectionistic strivings, however, reduced the positive relationship between perceived control and life satisfaction. Perfectionistic concerns moderated the relationship between perceived control and life satisfaction—the higher the perfectionistic concerns, the weaker the positive relationship between perceived control and life satisfaction. The study found that individuals with high perfectionistic tendencies are not always dissatisfied with life, but that perfectionism weakens the relationship between basic psychological needs, perceived control, and life satisfaction. We argue that one way to improve happiness is by coaching individuals who are highly perfectionistic to become self-aware of their personality so both their perfectionistic strivings and concerns are more coherent with their values and goals or character.

## 1. Introduction

With the development of a richer economy and a better society, individuals have higher requirements and expectations for their quality of life. Thus, people pay more attention to their happiness [1]. Therefore, subjective well-being has increasingly become a topic discussed by researchers [2]. Subjective well-being includes emotional happiness and life satisfaction [3]. Among them, life satisfaction is a cognitive evaluation of one’s own life as a whole [2], which is of great significance for individual mental health [1]. In this context, besides factors such as basic psychological needs and perceived control [4,5,6], personality traits also affect individual life satisfaction [7]. For example, trait perfectionism is an important factor affecting life satisfaction [8,9]. Therefore, our aim is to investigate these relationships closer in order to provide some enlightenment for promoting people’s happiness.

### 1.1. Basic Psychological Needs and Life Satisfaction

Basic psychological needs are important factors that affect individual happiness. According to self-determination theory, the basic psychological needs of individuals can be divided into three aspects: autonomy, relatedness, and competence. The need for autonomy refers to the need to experience a sense of volition and choice in one’s activities. The need for relatedness reflects the need to feel loved and cared for by significant others [4,6]. The need for competence reflects the need to feel effective in one’s actions and to be able to achieve one’s goals. The satisfaction of basic psychological needs is conducive to social development and personal well-being [4,6]. In addition, current studies have found that the satisfaction of basic psychological needs is conducive to individual happiness and satisfaction [4,6,10,11,12]. The satisfaction of basic psychological needs will increase life satisfaction, while the frustration of basic psychological needs will decrease life satisfaction. The satisfaction of the three basic psychological needs provides psychological “nutrition” for individual psychological growth and life satisfaction, so the frustration of the three basic psychological needs will reduce individual life satisfaction. For example, if individuals can freely choose what they like to do, their autonomy needs will be satisfied, thus further promoting life satisfaction. When individuals feel that they can control the surrounding environment and the outcome of events, their competence needs will be satisfied, so as to obtain higher life satisfaction. When the individual’s relatedness needs are met, the individual will think that they have a relatively good and close interpersonal relationship, so the individual’s life satisfaction is further improved [12,13]. This indicates that the satisfaction of basic psychological needs can further promote the life satisfaction of individuals.

### 1.2. Perceived Control and Life Satisfaction

Perceived control is also an important variable affecting individual happiness. Perceived control, which is not a basic psychological need, broadly refers to an individual’s belief in his or her ability to influence his or her living environment [13]. Ryan and Deci argued that human beings have an innate tendency to achieve personal growth and assimilation by internalizing behavior into the self [14]. The process of internalization can lead to the autonomous regulation of behavior, that is, the behavior is fully integrated into the self, or a more controlled form of motivational regulation, that is, the behavior is only partially integrated into the self. This theory hypothesized that more autonomous motivational regulation leads to better psychological regulation and well-being, while more control motivational regulation leads to poorer psychological regulation and morbidity [15]. Therefore, the stronger the individual’s motivation to control, the lower their life satisfaction may be.

However, along with the process of socialization, individuals must learn to exert control over the environment by choosing behaviors that are conducive to achieving desirable outcomes and avoiding undesirable outcomes [16]. Therefore, control is also an important means for individuals to achieve happiness. Among them, there seems to be a certain contradiction between the negative influence of control motivation and the positive influence of control, and perceived control is the key concept to solve this contradiction because individuals may also feel happy when they think they have the ability to exercise control, albeit with great motivation to control [17,18,19]. For example, a study with a large sample showed that perceived control and life satisfaction showed a significant positive correlation [5]. Therefore, although control motivation has a negative significance for the physical and mental health of individuals, if individuals can have high perceived control, they will also experience high life satisfaction.

### 1.3. The Moderating Effect of Perfectionism

Importantly, the development of perfectionism is closely related to the early satisfaction of basic psychological needs. In the early stage, if an individual receives high autonomy support from parents, basic psychological needs will be satisfied and high autonomy motivation will be developed, which further promotes the emergence of perfectionistic strivings [20,21]. However, if parents adopt psychological control, the basic psychological needs of individuals will be frustrated and control motivation will be further developed, which leads to the emergence of perfectionistic concerns [22,23]. High control motivation, not autonomous motivation, is positively correlated with perfectionistic concerns [24,25,26].

Perfectionism refers to the multidimensional concept of personality, which sets extremely high standards. There are many models of perfectionism [8,9]. For example, Frost et al. developed a widely used measure of perfectionism called the Multidimensional Perfectionism Scale. They divided perfectionism into six dimensions, including personal standards, concerns over mistakes, parental expectations, parental criticism, organization, and doubts about actions [8,27]. Furthermore, Hewitt and Flett categorized perfectionism into three groups: self-oriented perfectionism, socially prescribed perfectionism, and other-oriented perfectionism [28]. Although these models have been widely used, factor analysis studies examined the basic structure of multidimensional perfectionism measures and supported two higher-order perfectionism dimensions commonly used in contemporary research, mainly including perfectionistic strivings and perfectionistic concerns [8,9,29,30,31]. What is more, the dimensions in each model can be divided into perfectionistic strivings and perfectionistic concerns [29,30,31,32]. Therefore, we choose perfectionistic strivings and perfectionistic concerns to measure perfectionism. Perfectionistic strivings reflect a self-oriented striving for perfection and exceedingly high personal standards of performance, which is the forms, aspects, and subordinate dimensions of perfectionism. In contrast, perfectionistic concerns reflect concerns over making mistakes, fear of negative social evaluation if not perfect, doubts about actions, feelings of discrepancy between one’s high standards and actual performance, and negative reactions to imperfection, which is the forms, aspects, and subdimensions of perfectionism [31,33,34].

Dissatisfaction is considered a characteristic of high perfectionism because perfectionists want to be perfect in all domains of their lives, that is, they are always dissatisfied with one or more life domains [33]. Although previous studies have investigated the relationship between perfectionism and life satisfaction [34,35], few studies have investigated the role of perfectionism in the relationship between life satisfaction and factors, such as basic psychological needs and perceived control. Previous studies have found that life satisfaction is significantly positively correlated with perfectionistic strivings, and negatively correlated with perfectionistic concerns [33,34].

Perfectionism moderates the effects of various factors on life satisfaction. Individuals with perfectionism, including perfectionistic strivings and perfectionistic concerns, have extremely high standards [33,34]. These extremely high standards lead to two conditions. On the one hand, what satisfies non-perfectionists may not satisfy perfectionists. On the other hand, the same amount of increase in basic psychological needs and perceived control may induce different amounts of increase in life satisfaction for non-perfectionists and perfectionists. This is because extremely high standards not only mean that a higher level of the factor is required to satisfy the individual’s life, but also that a higher increase in the factor is required to achieve the same increase in life satisfaction. It is thus reasonable to expect that perfectionism may moderate the relationship between basic psychological needs, perceived control, and life satisfaction.

In adult life, individuals with high perfectionistic strivings have higher life satisfaction and less desire for basic psychological needs than those with low perfectionistic strivings due to the early satisfaction of basic psychological needs. It can be inferred that, compared with those with low perfectionistic strivings, individuals with high perfectionistic strivings generally have higher life satisfaction, and the positive relationship between basic psychological needs and life satisfaction will decrease. What is more, compared with those with low perfectionistic concerns, individuals with high perfectionistic concerns have lower life satisfaction due to the early frustration of basic psychological needs, and the positive relationship between basic psychological needs and life satisfaction will increase [22,23].

Moreover, we expect that perfectionism may moderate the relationship between perceived control and life satisfaction. Previous studies have shown that not only individuals with high perfectionistic concerns had control motivation, but also individuals with high perfectionistic strivings had certain control motivation [24,25,26]. Therefore, after individuals form perfectionism due to early environmental and internal interactions, the acquisition of perceived control becomes an important factor for their mental health [36,37].

Compared to non-perfectionists, perfectionists may employ maladaptive strategies to gain perceived control. For example, perfectionists attach too much importance to impossible goals, and when they fail to achieve the goal, they feel guilty, which leads to a decrease in individual happiness [38]. Due to the influence of the individual’s early environment or due to the requirements of parents, the individual may excessively increase the standard and think that the standard is of high value, which might cause various psychological problems [22,23]. Therefore, compared with non-perfectionists, perfectionists tend to adopt maladaptive control strategies because they attach too much importance to established standards, which can reduce the impact of perceived control on life satisfaction. However, non-perfectionists will adopt adaptive control strategies, that is, they will voluntarily give up when they feel they cannot achieve their goals and make self-biased attributions, which will increase the individual’s life satisfaction to a certain extent [36,38]. Therefore, perfectionism may moderate the relationship between perceived control and life satisfaction because perfectionists may employ maladaptive strategies to gain perceived control.

The following four hypotheses are proposed.

**H1:** 
*Perfectionistic strivings moderate the relationship between basic psychological needs and life satisfaction. More specifically, high perfectionistic strivings reduce the positive relationship between basic psychological needs and life satisfaction.*


**H2:** 
*Perfectionistic concerns moderate the relationship between basic psychological needs and life satisfaction. That is, high perfectionistic concerns enhance the positive relationship between basic psychological needs and life satisfaction.*


**H3:** 
*Perfectionistic strivings moderate the relationship between perceived control and life satisfaction. The higher the perfectionistic strivings, the lower the positive relationship between perceived control and life satisfaction.*


**H4:** 
*Perfectionistic concerns moderate the relationship between perceived control and life satisfaction. The higher perfectionistic concerns are, the lower the positive relationship between perceived control and life satisfaction will be.*


## 2. Materials and Methods

### 2.1. Procedures and Participants

An anonymous survey was administered via a free survey website (Wenjuanxing; https://www.wjx.cn, accessed on 3 September 2022). The students could only access the survey via a smartphone for one-time access and no identifying or tracking information was collected. The questionnaires of this study were presented online in the order of basic psychological needs, life satisfaction, perfectionism, and perceived control. The subjects completed the questionnaires in order in their spare time. Two validity items were designed, requiring the participants to choose a fixed option on the question. For example: for this question, please select “more consistent” [39]. If these two questions were not answered correctly, the questionnaire was judged invalid.

A total of 600 questionnaires were distributed at Huaibei Normal University, Xuzhou Medical University, and Chizhou University, and 590 questionnaires were collected. After removing the invalid questionnaires, 574 questionnaires were left, with a mean age of 19.53 (SD = 1.61), including 299 women and 275 men. According to the formula of sample size [40], the sample size of this paper should be greater than 267. Therefore, the sample size of this paper is sufficient. Participants were informed of the purpose of the study and their right to withdraw from the study at any stage. Participants were assured anonymity. The study was conducted in accordance with the Declaration of Helsinki and approved by the Institutional Review Board of Huaibei Normal University, its protocol code is HBNU-2022-6-15-05, and the date of approval is 15 June 2022. The results of the descriptive analysis are shown in Table 1.

### 2.2. Measures

#### 2.2.1. Perfectionism

In 2006, ZI and Zhou revised the Chinese version of the Multidimensional Perfectionism Scale [41]. This scale contains 27 items and includes five dimensions: personal standards, concerns over mistakes, parental expectations, doubts about actions, and organization. Participants were asked to rate each item on the 4-point Likert scale ranging from 1 (not meet) to 5 (meet). This study adopted the personal standards subscale, which represents perfectionistic strivings (6 items; e.g., “I set my sights higher than most people”), and the concerns over mistakes subscale, which represents perfectionistic concerns (6 items; “if someone works or studies better than me, I feel like a total failure”) [42]. The multidimensional perfectionism scale has been locally revised in China and has good reliability and validity [41]. The Mcdonald’s Omega coefficients of the personal standards subscale and concerns over mistakes subscale were 0.82 and 0.91, respectively.

#### 2.2.2. Basic Psychological Needs

The Basic Psychological Needs Scale was compiled by Deci and Ryan and is mainly used to assess the degree of satisfying the basic psychological needs of individuals in real life [43]. The Chinese version of the Basic Psychological Needs Scale was adopted in this study [44]. There are 21 items in this scale, which measure competence needs (6 items; e.g., “I often feel inadequate or incompetent”), relatedness needs (9 items; “I do like the people I interact with”), and autonomy needs (6 items; “I feel like I’m in charge of how I live my life, and I’m in charge of my life”). Participants were asked to rate each item on the 7-point Likert scale ranging from 1 (strongly disagree) to 7 (strongly agree). The Basic Psychological Needs Scale has been locally revised in China and has good reliability and validity [44]. Mcdonald’s Omega coefficient of basic psychological needs in this study was 0.88.

#### 2.2.3. Perceived Control

The Chinese version of the Sense of Control Scale revised by Li was adopted, including two dimensions of sense of control (6 items; e.g., “I can do almost anything I set my mind to”) and sense of restriction (6 items; “What I can and cannot do is largely determined by others”), with a total of 12 items and a 7-point score [45,46]. The higher the score, the higher the sense of control. The Sense of Control Scale has been locally revised in China and has good reliability and validity [45,46]. In this study, the Mcdonald’s Omega coefficient of this scale was 0.79.

#### 2.2.4. Life Satisfaction

The Life Satisfaction Scale was developed by Diener et al. [47,48]. The revised Chinese version has a total of 5 items (e.g., “I am satisfied with my life”) [49], and adopts a 7-point Likert scale. The higher the score, the higher the satisfaction with life. The Life Satisfaction Scale has been locally revised in China and has good reliability and validity [48]. In this study, the Mcdonald’s Omega coefficient of this scale was 0.88.

### 2.3. Statistical Analysis

SPSS 7.0 and PROCESS macro [50] were used for statistical analysis. First, SPSS 7.0 was used to evaluate the mean and standard deviation of each variable and the correlation of each variable. Then, PROCESS macro [50] was used to examine the moderating role of perfectionism. On the premise that the moderating effect is significant, the simple slope of the variable was further analyzed. That is, the moderating variables are divided into high (higher than the average plus one standard deviation) and low (lower than the average plus one standard deviation) groups for a simple slope test [51].

### 2.4. Common Method Bias

There may be a risk of common method bias in collecting data by questionnaire method. Therefore, this study followed previous methods to control common method bias (anonymous method and reverse scoring of some items), and uses the Harman single-factor test to test common method bias [52]. There are 7 factors with characteristic roots greater than 1. The first factor explains 22.51% of the total variation, which is less than the critical value of 40%, which confirms that there is no serious common method deviation in this study.

## 3. Results

### 3.1. Descriptive Statistics and Correlation Analysis

The results of descriptive statistics and correlation analysis are shown in Table 1. A total of 574 data (275 men and 299 women) were analyzed. Life satisfaction was positively correlated with perfectionistic strivings (*r* = 0.11, *p* < 0.01) and perceived control (*r* = 0.37, *p* < 0.001), and negatively correlated with perfectionistic concerns (*r* = −0.15, *p* < 0.01). Basic psychological needs were positively correlated with perceived control (*r* = 0.66, *p* < 0.001) and life satisfaction (*r* = 0.57, *p* < 0.001), and negatively correlated with perfectionistic concerns (*r* = −0.50, *p* < 0.001). Perceived control was negatively correlated with perfectionistic concerns (*r* = −0.46, *p* < 0.001).

### 3.2. The Moderating Effect of Perfectionistic Strivings

In this study, the PROCESS macro program [50] was used for data processing. Before formal data processing, all variables were standardized. On this basis, the moderating effect of perfectionism was tested. The research results are shown in Table 2. Perfectionistic strivings moderated the relationship between basic psychological needs and life satisfaction and the relationship between perceived control and life satisfaction.

The results showed that perfectionistic strivings moderated the relationship between basic psychological needs and life satisfaction, so we further conducted a simple slope analysis. The high perfectionistic strivings (higher than the average plus one standard deviation) and low perfectionistic strivings groups (lower than the average plus one standard deviation) were selected for the simple slope test [51]. High perfectionistic strivings group: simple slope = 0.14, 95% confidence interval [0.19, 0.26], *p* < 0.001. Low perfectionistic strivings group: simple slope = 0.22, 95% confidence interval [0.12, 0.18], *p* < 0.001. The simple slope is shown in Figure 1. This suggests that basic psychological needs can increase life satisfaction, but perfectionistic strivings reduce the positive relationship between basic psychological needs and life satisfaction. At the same time, the life satisfaction of those with perfectionistic strivings was generally higher, and regardless of the level of basic psychological needs, the life satisfaction of individuals with high perfectionistic strivings was higher than that of those with low perfectionistic strivings.

The results showed that perfectionistic strivings moderated the relationship between perceived control and life satisfaction, so a simple slope analysis was conducted. High perfectionistic strivings group: simple slope = 0.17, 95% confidence interval [0.10, 0.35], *p* < 0.001. Low perfectionistic strivings group: simple slope = 0.27, 95% confidence interval [0.22, 0.35], *p* < 0.001. The simple slope is shown in Figure 2. This suggests that the increase in perceived control will increase life satisfaction, but high perfectionistic strivings will reduce the positive relationship between perceived control and life satisfaction. At the same time, the life satisfaction of those with perfectionistic strivings was generally higher, and regardless of the level of basic psychological needs, the life satisfaction of individuals with high perfectionistic strivings was higher than that of those with low perfectionistic strivings.

### 3.3. The Moderating Effect of Perfectionistic Concerns

The results showed that perfectionistic concerns did not moderate the relationship between basic psychological needs and life satisfaction, but moderated the relationship between perceived control and life satisfaction. Simple slope analysis further analyzed the moderating effect of perfectionistic concerns on the relationship between perceived control and life satisfaction. Additionally, the high perfectionistic concerns group (higher than the average plus one standard deviation) and the low perfectionistic concerns group (lower than the average plus one standard deviation) were selected for the simple slope test [51]. High perfectionistic concerns group: simple slope = 0.15, 95% confidence interval [0.06, 0.22], *p* < 0.001. Low perfectionistic concerns group: simple slope = 0.27, 95% confidence interval [0.21, 0.33], *p* < 0.001. The simple slope is shown in Figure 3. This suggests that increased feelings of control increase life satisfaction, but perfectionistic concerns reduce the positive relationship between perceived control and life satisfaction. At the same time, when the perceived control is low, individuals with high perfectionistic concerns have better life satisfaction. When perceived control is high, individuals with low perfectionistic concerns have better life satisfaction.

## 4. Discussion

This study confirmed that perfectionistic strivings could not only moderate the relationship between basic psychological needs and life satisfaction, but also the relationship between perceived control and life satisfaction. Perfectionistic concerns only moderated the relationship between perceived control and life satisfaction. At the same time, this study also shows that perfectionists are not dissatisfied with life. Additionally, perfectionistic strivings were positively correlated with life satisfaction, while perfectionistic concerns were negatively correlated with life satisfaction. The study found that while high perfectionism does not always lead to life dissatisfaction, it does reduce the positive association between positive factors and life satisfaction.

### 4.1. Perfectionistic Strivings Moderate the Relationship between Basic Psychological Needs and Life Satisfaction

This study supported H1, namely, perfectionistic strivings can moderate the relationship between basic psychological needs and life satisfaction. Although there is a significant positive correlation between basic psychological needs and life satisfaction regardless of the level of perfectionistic strivings, high perfectionistic strivings will reduce the positive relationship between basic psychological needs and life satisfaction. Previous studies have shown that perfectionistic strivings are closely related to the satisfaction of early psychological needs of individuals [20,21,53]. If individuals meet their basic psychological needs at an early stage, they will generally be more satisfied with life [25,26,28]. At the same time, because of the satisfaction of early psychological needs, individuals with perfectionistic strivings generally have higher life satisfaction, which is relatively stable in adulthood.

This study found that perfectionistic concerns do not moderate the relationship between basic psychological needs and life satisfaction, which contradicts H2. Previous studies have shown that individuals are more likely to have perfectionistic concerns when they are frustrated with their basic psychological needs at an early stage [22,23,53,54]. This study also found a significant negative correlation between perfectionistic concerns and basic psychological needs. Therefore, the reason why perfectionistic concerns cannot moderate the relationship between basic psychological needs and life satisfaction may be that when early basic psychological needs are frustrated or not met, individuals are more eager to meet basic psychological needs and are more sensitive to the satisfaction of basic psychological needs. They all reported greater happiness as their basic psychological needs were met. This further shows the importance of satisfying early basic psychological needs because whether the basic psychological needs are frustrated or just not satisfied will make individuals pay more attention to the basic psychological needs at an early age. Then, as an adult, the pursuit of basic psychological needs can lead to increase individual control motivation (as long as they are in control of themselves and the external environment, in order to better meet their basic psychological needs), which leads to various mental problems [22,23].

### 4.2. Perfectionistic Strivings Moderate the Relationship between Perceived Control and Life Satisfaction

The study supported H3, namely, perfectionistic strivings moderate the relationship between perceived control and life satisfaction. When the perfectionistic strivings were high, life satisfaction is generally higher, but with the increase of the perceived control, the increase in life satisfaction is smaller. Additionally, when perfectionism is low, life satisfaction was generally low but increased more as the perceived control increases. There may be two reasons for this: (1) compared with individuals with low perfectionistic strivings, individuals with high perfectionistic strivings are more inclined to internal control [55,56,57], most of their targets are internal, so the overall perceived control (including the perceived control pointing inward and outward) has a lower impact on their happiness in life. (2) Individuals with high perfectionism generally have higher self-efficacy and higher ability [8,9], and higher self-efficacy and higher ability make individuals more stable in the face of the increase or decrease of perceived control, so they are less prone to the increase of perceived control, leading to an increase in life satisfaction.

### 4.3. Perfectionistic Concerns Moderate the Relationship between Perceived Control and Life Satisfaction

Supporting H4, perfectionistic concerns moderated the relationship between perceived control and life satisfaction. Among them, life satisfaction increased gradually with increasing perceived control regardless of the level of perfectionistic concerns. However, when perfectionistic concerns were high, life satisfaction increased less as perceived control increased. In contrast to the moderating effect of perfectionistic strivings on the relationship between perceived control and life satisfaction, when perceived control was high, life satisfaction was greater with low perfectionistic concerns than with high perfectionistic concerns.

Individuals with high perfectionistic concerns are more likely to employ maladaptive control strategies [36,37,38], that is, when faced with impossible standards, individuals do not give up and attribute uncontrollable events or impossible goals to themselves, leading to more serious psychological problems. This control strategy of perfectionistic concerns may lead to an increased perceived control but a slower increase in life satisfaction.

At the same time, when the perceived control is low, the life satisfaction of high perfectionistic concerns was greater than that of low life satisfaction, which may be because individuals with high perfectionistic concerns will develop a control strategy to increase their happiness in the face of low perceived control, but this strategy will slow down the influence of the perceived control on happiness when perceived control increase. This further suggests that perfectionistic concerns, which are negatively correlated with life satisfaction, are a risk factor for life satisfaction, not only directly affecting life satisfaction but also reducing the positive relationship between perceived control and life satisfaction.

### 4.4. Limitations

This study is a cross-sectional study. Therefore, future longitudinal studies are needed to further confirm the conclusions of this study. In addition, this study only uses two subscales of perfectionism, including perfectionistic strivings and perfectionistic concerns, to study the moderating effect of perfectionism, but there are other very common measures of perfectionism [30]. More measures of perfectionism are needed in the future to validate the conclusions of this study. Finally, life satisfaction is only one of three subjective well-being constructs (the other two being affect and harmony in life) [1,2,3]. These three constructs interact [58] and are part of a whole; thus, only using life satisfaction does not give the whole picture.

## 5. Conclusions

The results of the current study suggest that perfectionistic strivings not only moderated the relationship between basic psychological needs and life satisfaction, but also the relationship between perceived control and life satisfaction. What is more, perfectionistic concerns only moderated the relationship between perceived control and life satisfaction. This further shows the important role of trait perfectionism in well-being. It is not that perfectionists are unsatisfied with their lives, but perfectionism reduces the positive relationship between basic psychological needs, perceived control, and life satisfaction. This further suggests that the pursuit of high standards may reduce individuals’ perception of happiness. This also provides some enlightenment for clinical practice or the science of well-being. In clinical practice, perfectionism not only affects various mental problems but also weakens the effect of various interventions on individual well-being due to its extremely high standard. According to the science of well-being, for example, happiness can be further promoted by lowering the act of letting go of struggles that often are not of much importance in our lives [59,60,61]. One way of working with this is by coaching individuals who are highly perfectionistic to become self-aware of their personality so both their perfectionistic strivings and concerns are morecoherent with their values and goals or character [62,63].

## Figures and Tables

**Figure 1 behavsci-12-00440-f001:**
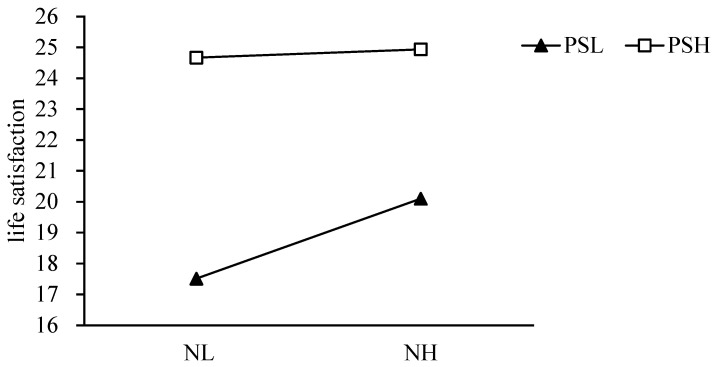
The moderating effect of perfectionistic strivings on the relationship between basic psychological needs and life satisfaction. NL: low basic psychological needs, NH: high basic psychological needs, PSL: low perfectionistic strivings, and PSH: high perfectionistic strivings.

**Figure 2 behavsci-12-00440-f002:**
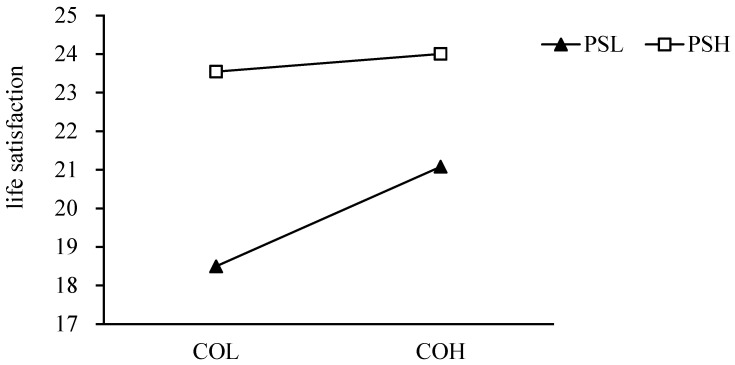
The moderating effect of perfectionistic strivings on the relationship between perceived control and life satisfaction. COL: low perceived control, NH: high low perceived control, PSL: low perfectionistic strivings, and PSH: high perfectionistic strivings.

**Figure 3 behavsci-12-00440-f003:**
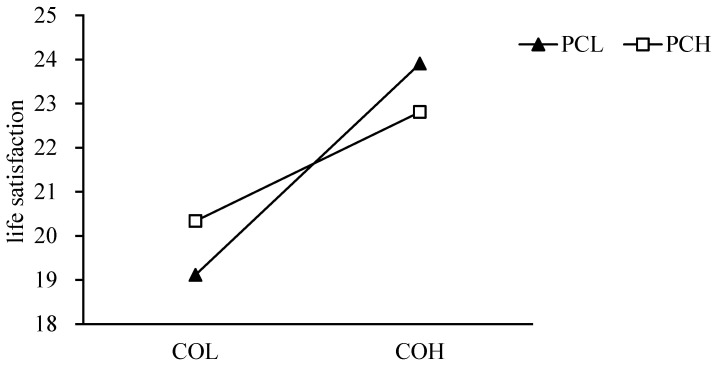
The moderating effect of perfectionistic concerns on the relationship between perceived control and life satisfaction. COL: low perceived control, NH: high low perceived control, PSL: low perfectionistic concerns, PSH: high perfectionistic concerns.

**Table 1 behavsci-12-00440-t001:** The results of descriptive statistics and correlation analysis.

	M	SD	1	2	3	4	5	Mcdonald’s Omega
1 Age	19.53	1.61	1					
2 Basic Needs	96.03	16.50	−0.16 **	1				0.88
3 Life Satisfaction	21.81	5.32	−0.10 *	0.57 ***	1			0.88
4 Perfectionistic Concerns	14.82	5.82	0.08	−0.50 ***	−0.15 **	1		0.91
5 Perfectionistic Strivings	18.62	4.68	−0.05	0.00	0.11 **	0.47 ***	1	0.82
6 Perceived Control	50.97	8.79	−0.14 **	0.66 ***	0.37 ***	−0.46 ***	−0.05	0.79

* *p* < 0.05, ** *p* < 0.01, *** *p* < 0.001.

**Table 2 behavsci-12-00440-t002:** Results of moderating analysis of life satisfaction.

Predictors	R	R^2^	F	*β*	Bootstrap Down	Bootstrap Up	*t*
Moderator: PS							
Basic Needs	0.60	0.36	104.96 ***	0.18	0.16	0.20	16.68 ***
Perfectionistic Strivings				0.15	0.08	0.23	−3.95 ***
PS × NE				−0.01	−0.01	0.00	−3.37 ***
Perceived Control	0.41	0.17	37.75 ***	0.23	0.18	0.27	9.77 ***
Perfectionistic Strivings				0.16	0.08	0.25	3.71 ***
PS × CO				−0.01	−0.02	0.00	−2.58 **
Moderator: PC							
Basic Needs	0.60	0.36	104.55 ***	0.21	0.18	0.23	16.06 ***
Perfectionistic Concerns				0.16	0.09	0.23	4.53 ***
PC × NE				0.00	−0.01	−0.00	−1.41
Perfectionistic Concerns	0.39	0.15	33.67 ***	0.21	0.15	0.26	7.44 ***
Perceived Control				0.01	−0.07	0.08	0.14
PC × CO				−0.01	−0.02	−0.00	−2.76 **

** *p* < 0.01, *** *p* < 0.001. NE: basic psychological needs, PC: perfectionistic concerns, PS: perfectionistic strivings, CO: perceived control.

## Data Availability

The data presented in this study are available on request from the corresponding author (email: zhaoxiaoyun1980@163.com).

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
