# Peer review of "Are Perfectionists Always Dissatisfied with Life? An Empirical Study from the Perspective of Self-Determination Theory and Perceived Control"

_behavsci, 2022, doi:10.3390/bs12110440_

Round 1
Reviewer 1 Report
Are Perfectionists Always Dissatisfied with Life? An Empirical Study from The Perspective of Self-Determination Theory and Perceived Control
The manuscript (ms) presents a study examining the moderating role of perfectionism in the relationship between perceived control and life satisfaction in a sample of college students. These variables are of interest to the readers of Behavioural Sciences journal. The findings are interesting, and the sample is of a sufficient size for the analyses. This being said, however, I have several comments and suggestions that would need to be addressed before I can recommend publication. These are outlined below.
Major comments/suggestions
The abstract requires work. Firstly, the abstract would benefit from a sentence or two to provide a rationale for the study. Secondly, more information on the participants is required (e.g., M age, SD). Thirdly, the aims of the study could be clearer. Fourthly, while a sentence indicates that implications for research and practice are discussed, it would be better to provide a brief overview of such implications.
A significant amount of work is required to provide a sufficient rationale for examining perfectionism dimensions as moderators of the relationships. A more comprehensive rationale for the study would enhance the introduction. In addition, the aims of the study could be clearer.
Perfectionism was conceptualised in a specific way in the present study. Perfectionism was conceptualised in a specific way in the present study. Numerous other models exist. These need to be discussed. Why did the researchers adopt their approach? The author do not provide a clear define perfectionism and there is a lack of justification for choosing this way of measuring perfectionism. The definitions of perfectionism dimensions (e.g., PS and PC) must be clearer. The authors also refer to PS as positive perfectionism. This is the a priori labelling of constructs as good or bad (adaptive/maladaptive or positive/negative). No psychological construct is inherently good or bad 100% of the time. By using such labelling, we are presuming to know the answer before we have asked the question, thus, rendering the scientific process obsolete. Perfectionistic strivings is an ambivalent form of perfectionism that usually shows positive and negative associations with adaptive and maladaptive consequences (see Hill & Curran, 2016). I feel this could be better explained in the ms. The point I am trying to make is that the authors need to carefully consider (a) the general classification and labelling of perfectionism dimensions and (b) providing a sound justification of adopting their approach. It will also be important to further acknowledge the overlap between the dimensions of perfectionism and the implications for your analyses (and the discussion of results; see Hill., 2014).
The terminology for the perfectionism dimensions changes throughout the ms. That is, in some instances, the authors use perfectionistic strivings and perfectionistic concerns and other times, they use perfectionist strivings and perfectionist concerns. Check throughout the ms to ensure the perfectionism variables are named in a consistent manner (perfectionistic strivings and perfectionistic concerns). I also suggest being consistent with terminology for life satisfaction. For example, happiness is used, however, happiness is not measured. It will be easier for the reader if you keep the terminology consistent with what is measured.
Please include a justification for sample size. I would prefer to see information on the participants of the study before the procedures and more details. That is, what was the final sample? What was the mean age? SD? I know this information is in the table, but I think it would make the participant section clearer.
Please include details of missing data and how this was dealt with. Any preliminary check on missing data/value and outliers? Any how did you deal with them? Provide detail on how the data were dealt with (e.g., missing data, skewness, kurtosis, univariate and multivariate outliers, normality).
While I have not used the sense of control scale, it seems to me that the dimensions reflect different and very contrasting types of control. I therefore suggest that the introduction is adapted to discuss the different dimensions, and the analyses are conducted with the two dimensions. This is also similar for the basic psychological needs (three dimensions) so I therefore suggest that the authors should provide rationale for analysing the dimensions together or re-analyse with separate dimensions.
Please replace Cronbach's alpha with Mcdonald's Omega and report in table 1.
Please provide support for validity and reliability of the scales.
Please add the size of the correlations. A claim is made about the relationship between perfectionistic strivings and life satisfaction; however, this is a weak correlation. Please check the significance level of the correlation.
The manuscript should be checked for missing references, especially for definitions (e.g., basic psychological needs, perfectionism) and measures. The manuscript could also be improved in terms of clarity. Please read through the paper and make grammatical changes and ensure there is a consistency in the language used for key variables throughout.
References
Hill, A. P. (2014). Perfectionistic strivings and the perils of partialling. International Journal of Sport and Exercise Psychology, 12, 302-315.
Hill, A.P., & Curran, T. (2016). Multidimensional perfectionism and burnout: A meta-analysis. Personality and Social Psychology Review, 20(3), 269–288.
Reviewer 2 Report
The article deals with an interesting and innovative topics. The methodology is quite detailed. The study group is quite large. The authors clearly present the results of their research. Discussion conducted correctly.
I only have comments on the references - references numbers: 3; 16; 36 they are from distant years. Please delete them or replace them with newer ones.
Reviewer 3 Report
Section 4.4: "future research is needed to study whether various types of perfectionism moderate all three basic psychological needs". Since data have been collected for these three basic psychological needs, why not analyze using the existing data and report it?
Round 2
Reviewer 3 Report
Although I think the further analysis would add more value to their paper, I accept the authors' explanations.
However, considering that they already measured the three basic psychological needs in their study, I don't agree with them still suggesting the need for future research on the three basic psychological needs in the Limitations section. It seems inappropriate.
Besides this concern, I do not have any more comments.
Author Response
Thank you for your understanding. In the future, we will study perfectionism and basic needs refer to your comments. As requested, we took away the the sentence in the Limitations section.